# An Insight into the Impact of Serum Tellurium, Thallium, Osmium and Antimony on the Antioxidant/Redox Status of PCOS Patients: A Comprehensive Study

**DOI:** 10.3390/ijms24032596

**Published:** 2023-01-30

**Authors:** Manal Abudawood, Lulu Alnuaim, Hajera Tabassum, Hazem K. Ghneim, Mohammad A. Alfhili, Samyah T. Alanazi, Naif D. Alenzi, Sarah Alsobaie

**Affiliations:** 1Department of Clinical Laboratory Sciences, College of Applied Medical Sciences, King Saud University, Riyadh 11362, Saudi Arabia; 2Department of Obstetrics and Gynaecology, Reproductive Medicine and Endocrinology, College of Medicine, King Saud University, Riyadh 11472, Saudi Arabia; 3Research and Laboratories Sector, National Drug and Cosmetic Control Laboratories (NDCCL), Saudi Food and Drug Authority, Riyadh 13513, Saudi Arabia

**Keywords:** tellurium, thallium, antimony, osmium, total antioxidant capacity, redox status, polycystic ovarian syndrome (PCOS)

## Abstract

Humans exploit heavy metals for various industrial and economic reasons. Although some heavy metals are essential for normal physiology, others such as Tellurium (Te), Thallium (TI), antimony (Sb), and Osmium (Os) are highly toxic and can lead to Polycystic Ovarian Syndrome (PCOS), a common female factor of infertility. The current study was undertaken to determine levels of the heavy metals TI, Te, Sb and Os in serum of PCOS females (*n* = 50) compared to healthy non-PCOS controls (*n* = 56), and to relate such levels with Total Antioxidant Capacity (TAC), activity of key antioxidant enzymes, oxidative stress marker levels and redox status. PCOS serum samples demonstrated significantly higher levels of TI, Te, Sb and Os and diminished TAC compared to control (*p* < 0.001). Furthermore, there was significant inhibition of SOD, CAT and several glutathione-related enzyme activities in sera of PCOS patients with concurrent elevations in superoxide anions, hydrogen and lipid peroxides, and protein carbonyls, along with disrupted glutathione homeostasis compared to those of controls (*p* < 0.001 for all parameters). Additionally, a significant negative correlation was found between the elevated levels of heavy metals and TAC, indicative of the role of metal-induced oxidative stress as a prominent phenomenon associated with the pathophysiology of the underlying PCOS. Data obtained in the study suggest toxic metals as risk factors causing PCOS, and thus protective measures should be considered to minimize exposure to prevent such reproductive anomalies.

## 1. Introduction

Rapid industrialization, urbanization, use of domestic/agricultural metallic compounds, and other anthropometric activities across the world have resulted in an undesirable impact on public health, mainly due to pollution from toxic heavy metals [1,2]. Women exposed to exogenous environmental heavy metals are susceptible to health risks since minute exposure to such toxic metals may give rise to reproductive dysfunction [3,4]. Exposure can occur via contaminated water, food, and soil contact. Toxic metals can be detrimental to pregnant women and those at reproductive age by inducing hormonal changes, altering menstrual cycles, ovulation, and female fertility [5,6]. Heavy metals and non-essential metals belong to a class of Endocrine Disruptor elements (EDE) and are considered reproductive toxicants [7]. Amongst the serious reproductive anomalies, Polycystic Ovarian Syndrome (PCOS) has emerged as one of the leading causes of infertility, with a global prevalence of about 4 to 21% [8]. PCOS is an endocrine disorder characterized by chronic anovulation, menstrual abnormalities, hyperandrogenism, hyperinsulinemia and polycystic ovaries [9,10]. Regardless of intensive and systematic investigations in this field, most PCOS cases are obscure with unidentified causes. Currently, metal ions are considered to be a possible risk and an etiological factor underlying PCOS, and are gaining interest. Variation in the serum levels of heavy metals has been linked to PCOS [11,12,13], but with unresolved etiologies. However, several studies have demonstrated oxidative stress as a major causative factor of the toxicity and carcinogenicity of these toxicants [14,15,16,17]. Total antioxidant capacity (TAC), antioxidant activity and oxidative stress marker levels reflect the extent of oxidative damage in a biological system. TAC reflects the ability of cells to quench free radicals, thus providing protection from the detrimental effects of these radicals. TAC is affected by the alteration in levels of free radicals or antioxidants in blood [18].

Among the toxic heavy metals, tellurium (Te), thallium (TI), and osmium (Os) rank higher in toxicity than other metals as far as public health is concerned. However, surveying the previous literature, we found a lack of data reporting the toxicity of these heavy metals especially in PCOS. As technology advances, demands for the rare element Te are on the rise. It is highly toxic, and is known to induce both acute and chronic toxicity in a variety of organisms. Te is reported to be involved in neurotoxicity, demyelination, oxidative stress, and neuronal death [19].

Thallium is a rare element that can be more toxic to humans than mercury, cadmium, or lead. Due to its odorless, tasteless, and water-soluble characteristics, the non-occupational population is exposed to continuous low doses of thallium in daily life unknowingly through the consumption of contaminated food and water, or through the inhalation of polluted air [20,21,22]. Ghaderi et al., 2015 reported that the presence of TI in any amount in the body is considered as abnormal [23]. Although data on the effects of TI on human reproduction is inadequate, TI toxicity could adversely affect the menstrual cycle, libido and male potency. Extreme toxicity can lead to alopecia, blindness and even death in some cases. Another important element that can be pro-toxic to humans is osmium (Os). Chronic exposure to osmium compounds can result in its accumulation in the kidney and liver causing organ damage [24]. Osmium tetroxide has been reported to cause reproductive toxicity in animals, but results related to human and developmental toxicity are ambiguous. Antimony (Sb) is an additional metal which has industrial applications. Inhalation/exposure to antimony trisulfide dust has been shown to cause degenerative changes in the myocardium and related electrocardiographic abnormalities in a variety of animal species [25].

There is limited data on the role of the toxicity of currently investigated heavy metals levels in females with PCOS. Few or no studies have evaluated the metal’s serum levels in females with PCOS. However, recently the urinary levels of a few were reported in females of reproductive age [26]. Evidence and knowledge on certain heavy metal involvement in PCOS development still remain incomplete and limited. To the best of our knowledge, this is the first study investigating serum levels of tellurium, thallium, and osmium in PCOS patients, and can offer insights into the treatment and prevention of the disorder. Therefore, the current study aimed to determine the serum levels of Te, TI, Os and Sb, and evaluate the correlation between such levels and the antioxidant, pro-oxidant and redox status in PCOS patients relative to fertile controls.

## 2. Results

Of the total women participants of the study, 47.1% were controls and 52.8% were women with PCOS. Among PCOS women, 56% had irregular menses and 60% had acne (*p* < 0.01). Elevated levels of luteinizing hormone (LH) and triglycerides (TG) were found significant in PCOS group compared against controls at *p* < 0.001 and *p* < 0.05, respectively.

Table 1 data shows the Te, TI, Os and Sb serum levels among the study groups. Elevated Te, TI, Os and Sb levels in the PCOS group were statistically significant at *p* < 0.001 compared to the control group. In contrast, serum TAC was found to decrease significantly in PCOS patients compared to controls (*p* < 0.001).

Inter-element analysis demonstrated significant positive correlation among heavy metals in PCOS patients. Pearson coefficient (*r*) between Te, TI, Os, Sb and TAC are shown in Table 2. Serum Te levels in PCOS patients exhibited a significant positive correlation with TI, Sb and Os, and a significant negative correlation with TAC. Parallel to Te, there was a significant positive correlation between TI & Sb, Sb and Os, and a negative correlation between TI and TAC, Sb and TAC, and Os and TAC. However, the correlation between TI and Os was non-significant. Meanwhile, data from the Pearson correlation clearly showed the antagonistic involvement of heavy metals and TAC in PCOS.

Multiple regression plots depicting the negative correlation between heavy metals and TAC are illustrated in Figure 1 and Figure 2. In an attempt to acquire a detailed mechanistic insight into the observed oxidative injury associated with heavy metal exposure in PCOS patients, we performed a series of experiments to probe alterations in serum oxidative stress markers. As shown in Figure 3a, SOD activity significantly decreased in PCOS patients compared to controls from 9.91 ± 0.48 to 6.48 ± 0.12 µmol/min/mL. A similar decrease in CAT activity was also noted in Figure 3b (49.77 ± 2.47 to 35.96 ± 0.61 µmol/min/mL). Consequently, significant elevations in SOA and H_2_O_2_ levels were detected, increasing from 54.31 ± 0.86 to 75.62 ± 1.20 nmol/mL (Figure 3c) and from 12.13 ± 0.12 to 17.20 ± 0.18 nmol/mL (Figure 3d), respectively.

Next, we examined the levels of LPO and PCC in the sera of control and PCOS subjects. Figure 4a,b shows significant increases in both LPO (0.67 ± 0.03 to 0.95 ± 0.02 nmol/mL) and PCC (63.71 ± 3.12 to 109.6 ± 1.45 pmol/mL) of PCOS patients in comparison to their normal counterparts. We also examined possible dysregulation of glutathione status in PCOS patients. Compared to normal subjects, we observed significant inhibition of enzyme activities related to glutathione metabolism, namely, GPx (Figure 5a), GR (Figure 5b), GST (Figure 5c), GCL (Figure 5d), and GS (Figure 5e). Moreover, PCOS subjects had significantly depleted GSH stores (Figure 5f) and significant accumulation of GSSG (Figure 5g), with a disrupted GSH/GSSG ratio (Figure 5h).

## 3. Discussion

With an alarming increase in metal exposure and its associated risk on humans, the current study was undertaken to examine the potential causal relationship between some rare heavy metals such as Te, TI, Os and Sb in PCOS patients, and to assess their impact on the generation of oxidative stress markers, antioxidants and redox status. Research on these rare elements in PCOS is scarce, and a cause-effect relationship remains unexplored and unclear. To fill the gap, we assessed levels of Te, TI, Os and Sb in serum of PCOS women using ICP/MS. To the best of our knowledge, this is the first comprehensive report of its kind where a large number of parameters related to antioxidant capacity have been studied. The main hallmark of the current investigation was the significantly increased serum levels of TI, Te, Os and Sb in PCOS patients compared to controls (*p* < 0.001), with subsequent decreases in the activities of key antioxidant enzyme activities, increases in the levels of oxidative stress markers and a disturbed redox balance in favor of oxidation, all leading to significant lowering of TAC. In addition, results showed significant negative correlation between serum levels of the investigated heavy metals with TAC in PCOS samples. The elevated levels of the examined metals in these patients suggest significant exposure to these heavy metals. Additionally, the diminished TAC is indicative of considerable disturbance of the antioxidant system that might have resulted from metal intoxication. The outcome of the current study supports the hypothesis of the involvement of increased generation of reactive oxygen species leading to oxidative stress and redox imbalance as prominent key factors in the etiology of the underlying PCOS.

PCOS is one of the reproductive anomalies with varied clinical presentation. The key feature of this endocrinopathy is hyperandrogenism and several other reproductive disorders resulting from hormonal and metabolic alterations in these patients [9,10]. The pathophysiology of this endocrinopathy is still unclear; however, the heterogeneity of its features suggests that besides genetic factors, environmental factors are of scientific and clinical importance. Out of 106 females enrolled in the present study, there was no differences in the sociodemographic characteristics between the studied groups as reported in our previous study [11]. Nevertheless, 56% had irregular menses and about 60% had skin problems such as acne. These variables varied significantly among the studied groups. Furthermore, there was no significant correlation between BMI and the previously examined heavy metals.

As mentioned above, humans are exposed to heavy metals in everyday life, a phenomenon considered to be hazardous to human health. Hence, as a matter of serious concern over public health, and women’s health in particular, it is essential to focus on this issue and such exposures should be avoided. Recently, in a meta-analysis by Yin et al., 2020, higher levels of trace elements were reported in PCOS subjects [12]. Although association between PCOS and heavy metals has been researched all around the globe, the role of rare elements such as TI and Te is still unexplored. The current study emphasizes the causal correlation between the rare heavy metals TI, Te, Os, Sb, and TAC in PCOS. Data from ICP-MS revealed higher serum levels of TI, Te, Os and Sb in PCOS patients compared to controls, and the level of the TAC was observed to be lower in PCOS than controls.

Te is a rare element in the earth’s crust and occupies 16th position in the periodic table. It is a non-essential, toxic element, a metalloid mostly existing as an oxyanion (TeO_3_) [27], and has many industrial uses in electronics [28]. There are only a few reports on the toxicology of organotellurium compounds. Thus, concerns have been raised about potential environmental and human health issues, as some forms of Te are highly toxic and can lead to adverse effects on women, affecting reproduction and fertility outcomes. The role of Te in PCOS is still obscure. To unravel the association between Te and PCOS, the current study reported increased serum levels of Te, increased generation of reactive oxygen species, higher pro-oxidant levels, lowered antioxidant activity, and disturbed redox homeostasis in PCOS patients compared to controls, showing substantial toxicity of Te among these patients. The current positive correlation between elevated levels of these toxic elements and total antioxidant status reflects the deleterious action of these metals, involving oxidative damage, and thus shows the significance of the study. The decreased serum TAC in PCOS patients observed here is in accordance with the finding of Kanafchian et al., 2020 [19]. The most interesting biochemical action associated with tellurium compounds is the alteration of oxidant-antioxidant equilibrium in favor of oxidative stress via inhibition of essential antioxidant enzymes, with a subsequent increase of oxidative stress and cell death [29]. A study in Oman documented a similar association of oxidative stress with diminished TAC as a paramount feature underlying pathophysiology in PCOS [30].

TI is another heavy element with biological importance. TI is a cumulative poison and is linked to neurotoxicity and cardiovascular diseases. The Environmental Protection Agency (EPA), had listed it as a major pollutant, and it is on the list of priority hazardous substances [31]. It has been reported that Tl is more acutely toxic than Hg, Cd, Pb, Zn and Cu in mammals [32]. With regard to the general population, drinking water and eating food contaminated with Tl, mainly fish and shellfish, constitute a major source of exposure to this metal [33]. Large fractions of TI are released into the atmosphere through anthropometric sources. Major sources of man-made TI pollution include gaseous emissions from cement factories, coal burning power plants, and metal sewers. Ore leaching constitutes a major source of TI contamination in water [1]. High-technology and future-technology fields have led to an increasing demand for TI [34]. TI is absorbed through skin and mucous membranes, and accumulates in bones, the renal medulla, and in the central nervous system. To date, toxicological studies on metals such as TI, Te and Sb are rare, as these elements are often undetected. This could be due to poor analytical methods with low sensitivity. ICP-MS is a sensitive and reliable technique for detection of element. As per ATSDR, blood concentrations of less than 1 ppb are recorded in normal individuals. However, TI concentrations in blood greater than 10 ppb, and as high as 50 ppb, are reflective of significant exposure [35,36]. Recently, Zeng et al., 2019 reported a concentration of 0.05 ppb of TI in Chinese adults [13]. In parallel, Gong et al., 2021 reported a similar range of TI in pregnant female [5]. Ma et al., 2021 studied TI exposure on maternal blood pressure and hypertensive disorder complicating pregnancy [37]. More recently in a study in Saudi Arabia, enhanced levels of TI with other elements such as Te, and Os were reported in females with recurrent miscarriage [38]. Nonetheless, no studies on the potential toxicity of TI in PCOS have been done, and this lack of data indicates the requirement for further research. The present study reports serum levels of TI of 12.69 ppb in PCOS patients. The exact mechanisms that mediate TI toxicity are still poorly understood. The diminished TAC observed in the present study could be a consequence of higher levels of TI in PCOS. Elevated levels of TI in PCOS patients is alarming as TI greater than 10 ppb is reflective of significant exposure and associated risk in this group. There are various mechanisms through which heavy metals induce toxicity. Impaired glutathione metabolism and disruption of potassium-regulated homeostasis are a few of the toxic effects suggestive of TI toxicity. TI can disrupt cellular function either by mimicking the potassium ion and substituting potassium in the Na+/K+ -ATPase, or by reacting with thiol groups [39], leading, as shown in this study, to lowered serum GSH levels. Parallel to Te, TI is also recognized to trigger oxidative stress in cells by inhibiting a number of enzymatically catalyzed reactions, thus interfering with vital metabolic processes and cell equilibrium leading to generalized poisoning [39]. This can result in the accumulation of oxidant species, which could adversely affect different molecules and their related cellular processes, as evident from the currently observed decreased TAC and antioxidant enzyme activities, and the increased generation of reactive oxygen species and oxidative stress markers.

Sb is an important element, is a silvery white metal with numerous industrial applications, and is released to the environment as a result of many natural discharges. Commonly, people are exposed to Sb, which is present in the pentavalent form in serum and the trivalent form in red blood cells. An average intake of about 5 μg of Sb/day from food and water has been reported [25]. Recently, Komarova et al., 2021 investigated heavy metal levels in the blood of the adult population of Queensland, Australia where Sb levels were detected in plasma at a concentration of 4 µg/L [40]. Pneumoconiosis, chronic bronchitis, chronic emphysema, and respiratory irritation after chronic exposure to antimony trioxide and/or pentoxide dust have been recorded [41]. Sb as a pro-toxic element, as evidenced by a report in which women working at an antimony metallurgical plant exhibited increased incidence of spontaneous abortions and disturbances in menstruation compared to controls [42]. Oxidative damage due to Sb toxicity has been studied previously. The increased serum Sb levels and the decreased TAC and antioxidant enzyme activities and increased generation of oxidative stress markers in PCOS patients in our study are in line with previous reports [26,42].

Os is a toxic element requiring constant vigilance. Osmium oxide (OsO_4_), an oxidized product of osmium, is a strong oxidizing agent. A recent case report of Os exposure documented significant absorption of OsO_4_ through skin and eyes, causing irreversible blindness in the case of direct exposure [43,44]. Furthermore, chronic exposure to osmium tetroxide has been shown to cause liver and kidney damage as a result of its accumulation in these organs [24]. Although reproductive toxicity by osmium tetroxide has been reported in animals, reproductive or developmental toxicity in humans remains unexplained. The significantly increased Os levels noted in PCOS patients might be involved in metal-induced oxidative stress leading to a sequence of events in PCOS. Furthermore, the inter-element relationship among heavy metals in PCOS patients, as evident by Pearson’s correlation results, was markedly positive, suggesting a joint effect of these heavy metals in triggering decreased antioxidant activity in favor of oxidative stress, which could be recognized as risk factors in the PCOS etiology. The results of our study necessitate a rapid surveillance of these toxic metals on a larger scale, mainly to eliminate or minimize exposure risks among females and avoid deleterious effects on women’s reproductive health.

## 4. Materials and Methods

### 4.1. Ethical Approval and Study Design

This study was carried out in collaboration with the Department of Obstetrics and Gynecology, King Khalid University Hospital (KKUH), King Saud University Medical City, Riyadh from January 2019 to February 2021. Ethical approval was obtained from the ethical committee at King Saud University (E-18-3536), and patient consent was obtained after a face-to-face interview and completion of a questionnaire in which information related to lifestyle, demography, anthropometry, menstrual history, and possible exposure to heavy metals was acquired.

The current case-controlled study enrolled 106 subjects aged 19–35 years. Healthy, non-PCOS, fertile females asymptomatic of hyperandrogenism, infertility, menstrual dysfunction, and sonographic signs of PCOS were assigned to the control group. Participants assigned to the case group had at least two features of the following; (a) oligo- or amenorrhea (<eight menstrual cycles in the current year), (b) hyperandrogenism, or (c) polycystic ovaries. Screening of PCOS was based on the Rotterdam criteria [45]. Exclusion criteria included pregnancy, diabetes mellitus, anemia, malignant neoplasms, active infectious disease, thromboembolism, stroke, ischemic heart disease, and administration of lipid-lowering or antihypertensive drugs.

### 4.2. Collection of Sample and Measurement of Baseline Characteristics

Blood (5 mL) was drawn by venipuncture and centrifuged at 3000 rpm for 15 min. Sera were then aliquoted into Eppendorf tubes and stored at −80 °C until the time of analysis. A basic biochemistry profile was obtained for all samples using Cell Dyne 3700 chemistry analyzer (STA Compact, Mediserv, UK).

### 4.3. Serum Te, TI, Os, and Sb Determination

Serum levels of all investigated heavy metals were determined by inductively-coupled plasma mass spectrophotometry (ICP/MS) using Agilent 7700 Series (Agilent Technologies, Santa Clara, CA, USA). For analysis of elements-Te, TI, Os and Sb certified reference material (ISO-17034 certified Agilent standard) of each element was used. Quality assurance was obtained by running internal Q.C with CRM (certified reference material) before each element analysis.

A total of 400 µL of serum was centrifuged and diluted with 2.5 mL of solvent mix containing 1% *v*/*v* HNO_3_ and 0.01% *v*/*v* Triton X100. Measurements were obtained after running a parallel calibration curve with a detection range of 0.05–100 ppb prepared from a standard stock solution.

### 4.4. Assessment of Antioxidant/Redox Status in Serum

A total Antioxidant Capacity Assay Kit (Millipore-Sigma, Burlington, MA, USA) was used to measure the antioxidant status of serum samples. TAC was measured by monitoring the reduction of copper at 570 nm and was expressed in µmol/L.

#### 4.4.1. Superoxide Dismutase (SOD), Catalase (CAT), Superoxide Anion (SOA), and Hydrogen Peroxide (H_2_O_2_)

SOD was measured by the method of Haan et al., 1996 [46] using xanthine oxidase at λ_max_ = 530 nm as previously modified [47]. CAT activity was assayed using H_2_O_2_ breakdown at 280 nm using a Catalase Activity Assay Kit (Solarbio Life Sciences, Beijing, China) as previously described [48]. Serum levels of SOA were measured as a function of formazan blue formation (λ_max_ = 450 nm) as described previously [49], while H_2_O_2_ content was detected by Amplex Red (Thermo Fisher, Waltham, MA, USA) fluorescence at Ex/Em of 530/590 nm [50].

#### 4.4.2. Lipid Peroxidation (LPO) and Protein Carbonylation (PCC)

LPO was estimated from malondialdehyde levels [51], whereas PCC was derived by reacting sera with dinitrophenylhydrazine [52]. For LPO determination, briefly 100 µL of sera was added to trichloroacetic acid (1 mL, 17.5%) and thiobarbituric acid (1 mL, 0.6%) and incubated for 15 min at 100 °C. After cooling, trichloroacetic acid (1 mL, 70%) was then added to the mixture, it was then left standing at room temperature for 20 min and centrifuged at 2000 rpm for 15 min. The supernatant was removed and the absorbance measured at 535 nm to determine MDA concentration. Protein carbonylation content (PCC) was measured by reacting carbonylated proteins with dinitrophenylhydrazine as outlined by Reznick and Packer [52].

#### 4.4.3. Glutathione Status

Activities of glutathione peroxidase (GPx), glutathione reductase (GR), glutathione-S-transferase (GST), glutamate cysteine ligase (GCL), glutathione synthetase (GS), reduced glutathione (GSH), and oxidized glutathione (GSSG) were measured as previously reported [53,54].

### 4.5. Statistical Analysis

Sigma Plot was used for statistical analysis. The Shapiro-Wilk test was applied to test normality. Data were normally distributed. Comparisons of serum levels of Te, TI, Os, Sb, and those of the antioxidant markers, GSH and GSSG between control and PCOS subjects were performed by Student’s *t*-test. In order to investigate the role of heavy metals on total antioxidant status, Pearson’s correlation and multiple regression was applied between heavy metals and TAC. *p* values at <0.05 were considered statistically significant and *p* values at <0.001 were considered highly significant.

## 5. Conclusions

In light of the current study results, it is concluded that the toxic heavy metals TI, Te, Os and Sb can be recognized as potential toxicants in the etiology underlying PCOS. Diminished antioxidant status and concurrent increases in oxidants and pro-oxidants, as well as disruption of redox homeostasis, account for the observed oxidative damage. Heavy metal intoxication, together with aberrant oxidative balance, contributes substantially in the PCOS pathophysiology. It is thus essential to monitor Te, TI, Sb and Os levels in the environment and minimize exposure to avoid their detrimental effects on women’s health and reproduction.

## Figures and Tables

**Figure 1 ijms-24-02596-f001:**
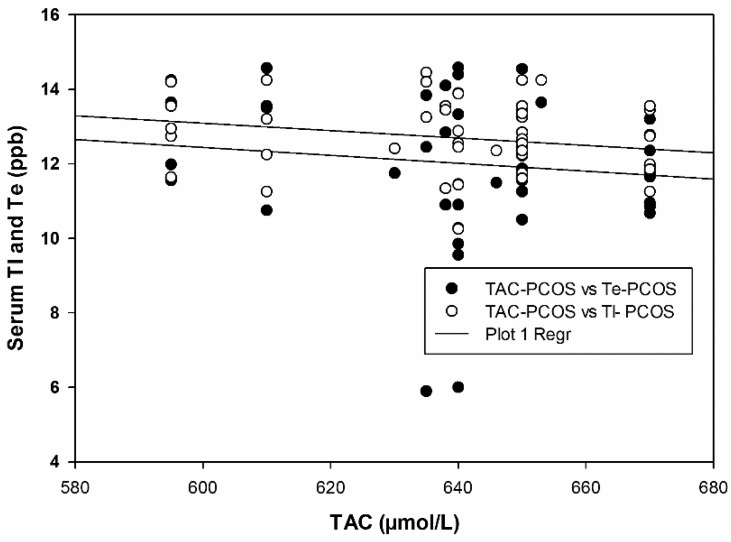
Multiple regression plot of TI and Te with TAC in PCOS.

**Figure 2 ijms-24-02596-f002:**
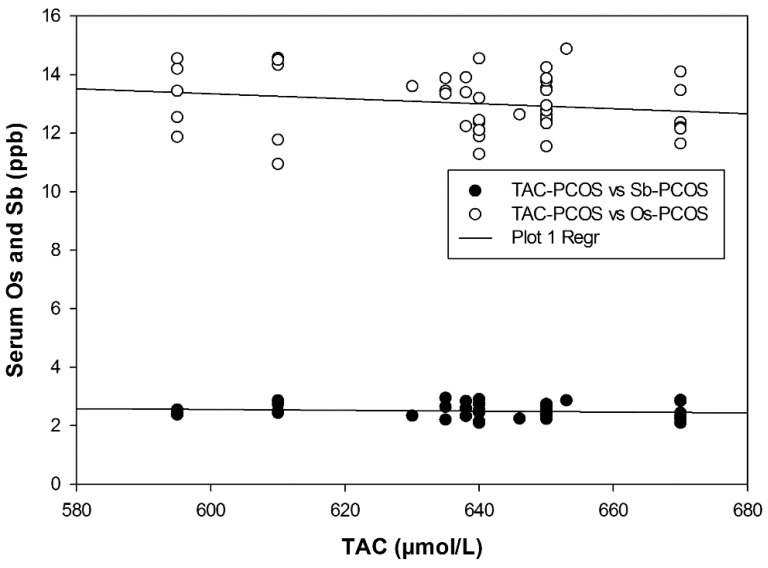
Multiple regression plot of Os and Sb with TAC in PCOS.

**Figure 3 ijms-24-02596-f003:**
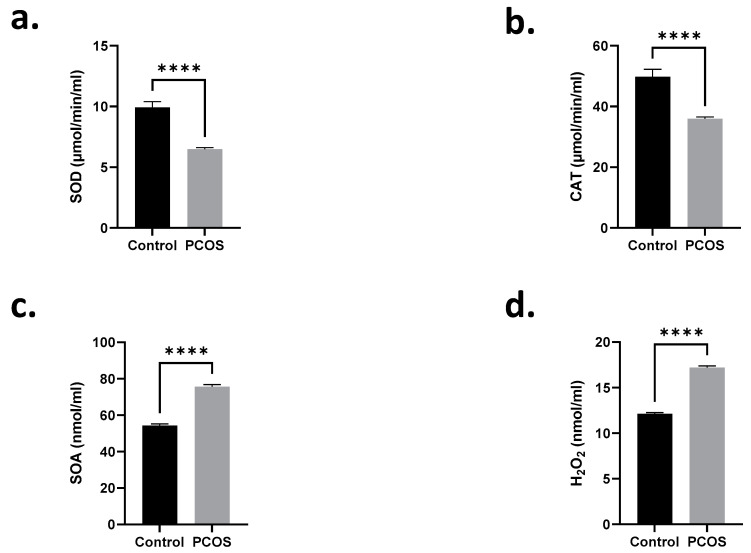
Serum SOD and CAT enzyme activities are inhibited in PCOS patients. Means ± SEM of (**a**) SOD and (**b**) CAT activities and (**c**) SOA and (**d**) H_2_O_2_ levels. **** *p* < 0.0001.

**Figure 4 ijms-24-02596-f004:**
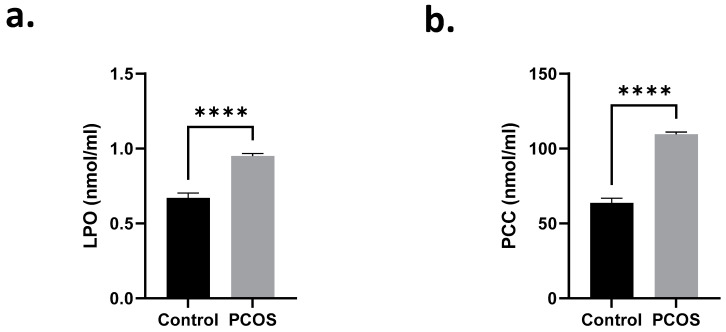
Serum LPO and PCC are elevated in PCOS patients. Means ± SEM of (**a**) LPO and (**b**) PCC levels. **** *p* < 0.0001.

**Figure 5 ijms-24-02596-f005:**
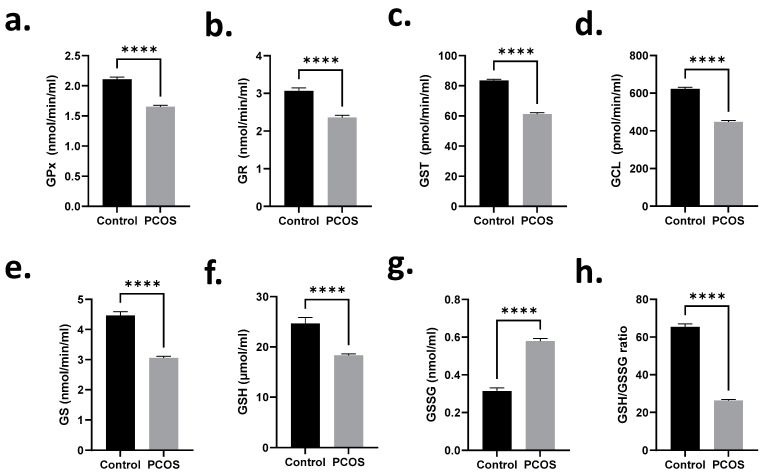
Glutathione homeostasis is disrupted in PCOS patients. Means + SEM of (**a**) GPx, (**b**) GR, (**c**) GST, (**d**) GCL, and (**e**) GS serum enzyme activities, levels of (**f**) GSH and (**g**) GSSG, and (**h**) GSH/GSSG ratio. **** *p* < 0.0001.

**Table 1 ijms-24-02596-t001:** Baseline characteristics and serum levels of heavy metals and TAC in controls and PCOS patients.

	Control	PCOS	*p*
Age	29.16 ± 6.2	30.41 ± 6.8	NS
BMI	25.0 ± 6.08	27.23 ± 5.0	NS
Married	30	24	
Thallium (ppb)	1.41 ± 0.4	12.69 ± 1.05	<0.001
Tellurium (ppb)	1.32 ± 0.46	12.33 ± 1.31	<0.001
Osmium (ppb)	1.51 ± 0.45	13.0 ± 0.97	<0.001
Antimony (ppb)	1.89 ± 0.31	2.50 ± 0.23	<0.001
TAC (µmol/L)	687.00 ± 11.80	640.00 ± 22.0	<0.001

NS—Non-significant.

**Table 2 ijms-24-02596-t002:** Correlation coefficients values Te, TI, Os, Sb and TAC in PCOS group.

	Te-PCOS *r* (*p*)	TI-PCOS *r* (*p*)	Sb-PCOS *r* (*p*)	Os-PCOS *r* (*p*)
TAC-PCOS	−0.132 (0.036) *	−0.210 (0.014) *	−0.142 (0.032) *	−0.194 (0.01) *
Te-PCOS		0.49 (<0.001) **	0.53 (<0.0001) ***	0.480 (<0.001) **
TI-PCOS			0.72 (<0.0001) ***	0.71 (NS)
Sb-PCOS				0.53 (<0.001) **

* indicates *p* < 0.05; ** *p* < 0.001;*** indicates *p* < 0.0001.

## Data Availability

The data sets generated during the current study are available from the corresponding author on reasonable request.

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
