# Peer review of "An Insight into the Impact of Serum Tellurium, Thallium, Osmium and Antimony on the Antioxidant/Redox Status of PCOS Patients: A Comprehensive Study"

_ijms, 2023, doi:10.3390/ijms24032596_

Round 1

Reviewer 1 Report

The manuscript entitled “An insight into the impact of serum Tellurium, Thallium, Osmium and Antimony on the antioxidant/redox status of PCOS patients: A comprehensive study” investigates Te, Tl Os and Sb in human serum of PCOS patients and investigation of the connection of these elements’ content in human serum of patients. The manuscript is within the aim and scope of the journal and after some minor revision I suggest it for publication.

The abstract and introductions are well and clear written. The aims should be clearly highlighted.

In the material and method of chemical analysis of Te, Tl, Os, Sb should be written details about quality control and assurance. The authors should write have they used some certified reference material.

The results and discussion are informative presented and the main conclusions were highlighted.

Author Response

Thanks for the peer review and comments. The manuscript is edited as per the suggestions from the reviewer. All the changes are inserted with Track changes On in the manuscript.

The manuscript entitled “An insight into the impact of serum Tellurium, Thallium, Osmium and Antimony on the antioxidant/redox status of PCOS patients: A comprehensive study” investigates Te, Tl Os and Sb in human serum of PCOS patients and investigation of the connection of these elements’ content in human serum of patients. The manuscript is within the aim and scope of the journal and after some minor revision I suggest it for publication.

The abstract and introductions are well and clear written. The aims should be clearly highlighted.

The aim of the study is revised and highlighted in the manuscript as suggested.

In the material and method of chemical analysis of Te, Tl, Os, Sb should be written details about quality control and assurance. The authors should write have they used some certified reference material.

As advised, the information is added in the manuscript in Method for chemical Analysis

Reviewer 2 Report

Heavy metals or semi-metals are known for their highly dramatic toxicity especially in relatively small but regular dosages. The toxicity is partially caused by increased oxidative stress in the tissue that can be induced by these toxins. Therefore, the topic of this manuscript is important, timely and interesting.

The text shows a logic structure. The authors used adequate methods and research design. The experimental data support results, which are illustratively presented. Additionally, the results and data are properly discussed.

Nevertheless, I recommend adding a brief description of the presented method "2.4.2. Lipid peroxidation (LPO) and protein carbonylation (PCC) ".

Author Response

Thanks for the peer review and comments. The manuscript is edited as per the suggestions from the reviewer. All the changes are inserted with Track changes On in the manuscript.

Heavy metals or semi-metals are known for their highly dramatic toxicity especially in relatively small but regular dosages. The toxicity is partially caused by increased oxidative stress in the tissue that can be induced by these toxins. Therefore, the topic of this manuscript is important, timely and interesting.The text shows a logic structure. The authors used adequate methods and research design. The experimental data support results, which are illustratively presented. Additionally, the results and data are properly discussed.

Nevertheless, I recommend adding a brief description of the presented method "2.4.2. Lipid peroxidation (LPO) and protein carbonylation (PCC) ".

As per the suggestion, a brief description of the method for LPO and PCC is added in section 2.4.2

Reviewer 3 Report

I have several comments for the study by Abudawood and co-workers.

 1. Current study is very similar to previously published paper of the same authors (Abudawood M, Tabassum H, Alanazi AH, et al. Antioxidant Status in Relation to Heavy Metals Induced Oxidative Stress in Patients With Polycystic Ovarian Syndrome (PCOS). Research Square;2021. DOI: 10.21203/rs.3.rs-491791/v1.). It seems that same subjects were used (106 subjects aged 19-35 years, 47.1% controls and 52.8% women with PCOS) despite slight differences in study duration (from October 2018 to December 2020 (in already published study) vs January 2019 to February 2021 in this study).

 2. There is no need to recapitulate in the text results already presented in tables.

 3. I suggest the authors to be careful regarding correlations. By some definitions correlation coefficient ranging from 0 to 0.1 represent weak association, from 0.1 to 0.3 moderate while over 0.5 is strong correlation. Although statistically significant, the correlation between heavy metals and TAC obtained by the authors are to my opinion weak.

 4. Please check the references. It seems that the references in the text are not the same as in the Reference section. For example reference number 11 in the text (page 8, line 248) is the reference number 12 in the list.

5.  In the light of already published study, the conclusion of the present paper is not appropriate. Increase in serum levels of Tl, Te, Os and Sb in women with PCOS  are not solely responsible for decrease in TAC. Oxidative stress probably have significant role in etiology of PCOS, and intoxication with examined heavy metals can contribute, but increase in As, Cd, Pb and Hg as well.

Author Response

Thanks for the peer review and comments. The manuscript is edited as per the suggestions from the reviewer. Result and Conclusion are edited as advised.All the changes are inserted with Track changes On in the manuscript.

Response to reviewer comments

  1. Current study is very similar to previously published paper of the same authors (Abudawood M, Tabassum H, Alanazi AH, et al. Antioxidant Status in Relation to Heavy Metals Induced Oxidative Stress in Patients With Polycystic Ovarian Syndrome (PCOS). Research Square;2021. DOI: 10.21203/rs.3.rs-491791/v1.). It seems that same subjects were used (106 subjects aged 19-35 years, 47.1% controls and 52.8% women with PCOS) despite slight differences in study duration (from October 2018 to December 2020 (in already published study) vs January 2019 to February 2021 in this study).

 Yes, the current study is the extension of our previous work on the same patients. Heavy metals (Te, Tl, Os, Sb) were assayed in relation to antioxidant, pro-oxidant and redox status in PCOS patients relative to fertile controls.This was not investigated in previous study.

  1. There is no need to recapitulate in the text results already presented in tables.

 The result section is edited as per the suggestion

  1. I suggest the authors to be careful regarding correlations. By some definitions correlation coefficient ranging from 0 to 0.1 represent weak association, from 0.1 to 0.3 moderate while over 0.5 is strong correlation. Although statistically significant, the correlation between heavy metals and TAC obtained by the authors are to my opinion weak.

We acknowledge that the correlation between heavy metals and TAC observed is not strong as per the correlation coefficient r-value. However, a lower r-value reflecting mild associations is sometimes considered significant and need to be considered in medical sciences, especially in cases of diseases posing a health risk to human. 

  1. Please check the references. It seems that the references in the text are not the same as in the Reference section. For example reference number 11 in the text (page 8, line 248) is the reference number 12 in the list.

References are checked and corrected as advised

  1.  In the light of already published study, the conclusion of the present paper is not appropriate. Increase in serum levels of Tl, Te, Os and Sb in women with PCOS  are not solely responsible for decrease in TAC. Oxidative stress probably have significant role in etiology of PCOS, and intoxication with examined heavy metals can contribute, but increase in As, Cd, Pb and Hg as well.

The conclusion is revised as per the suggestion

Round 2

Reviewer 3 Report

I have no comments.